# On Fairness and Calibration

**Geoff Pleiss**,* **Manish Raghavan**,* **Felix Wu**, **Jon Kleinberg**, **Kilian Q. Weinberger**
Cornell University, Department of Computer Science
{geoff,manish,kleinber}@cs.cornell.edu,
{fw245,kwq4}@cornell.edu

## Abstract

The machine learning community has become increasingly concerned with the potential for bias and discrimination in predictive models. This has motivated a growing line of work on what it means for a classification procedure to be "fair." In this paper, we investigate the tension between minimizing error disparity across different population groups while maintaining calibrated probability estimates. We show that calibration is compatible only with a single error constraint (i.e. equal false-negatives rates across groups), and show that any algorithm that satisfies this relaxation is no better than randomizing a percentage of predictions for an existing classifier. These unsettling findings, which extend and generalize existing results, are empirically confirmed on several datasets.

## 1 Introduction

Recently, there has been growing concern about errors of machine learning algorithms in sensitive domains – including criminal justice, online advertising, and medical testing [33] – which may systematically discriminate against particular groups of people [2, 4, 8]. A recent high-profile example of these concerns was raised by the news organization ProPublica, who studied a risk-assessment tool that is widely used in the criminal justice system. This tool assigns to each criminal defendant an estimated probability that they will commit a future crime. ProPublica found that the risk estimates assigned to defendants who did not commit future crimes were on average higher among African-American defendants than Caucasian defendants [1]. This is a form of false-positive error, and in this case it disproportionately affected African-American defendants. To mitigate issues such as these, the machine learning community has proposed different frameworks that attempt to quantify fairness in classification [2, 4, 8, 19, 26, 34, 37]. A recent and particularly noteworthy framework is Equalized Odds [19] (also referred to as Disparate Mistreatment [37]),[1] which constrains classification algorithms such that no error type (false-positive or false-negative) disproportionately affects any population subgroup. This notion of non-discrimination is feasible in many settings, and researchers have developed tractable algorithms for achieving it [17, 19, 34, 37].

When risk tools are used in practice, a key goal is that they are *calibrated*: if we look at the set of people who receive a predicted probability of $p$, we would like a $p$ fraction of the members of this set to be positive instances of the classification problem [11]. Moreover, if we are concerned about fairness between two groups $G_1$ and $G_2$ (e.g. African-American defendants and white defendants) then we would like this calibration condition to hold simultaneously for the set of people within each of these groups as well [16]. Calibration is a crucial condition for risk tools in many settings. If a risk tool for evaluating defendants were not calibrated with respect to groups defined by race, for example, then a probability estimate of $p$ could carry different meaning for African-American and white defendants, and hence the tool would have the unintended and highly undesirable consequence of incentivizing judges to take race into account when interpreting its predictions. Despite the

importance of calibration as a property, our understanding of how it interacts with other fairness properties is limited. We know from recent work that, except in the most constrained cases, it is impossible to achieve calibration while also satisfying Equalized Odds [8, 26]. However, we do not know how best to achieve relaxations of these guarantees that are feasible in practice.

Our goal is to further investigate the relationship between calibration and error rates. We show that even if the Equalized Odds conditions are relaxed substantially – requiring only that weighted sums of the group error rates match – it is still problematic to also enforce calibration. We provide necessary and sufficient conditions under which this calibrated relaxation is feasible. When feasible, it has a unique optimal solution that can be achieved through post-processing of existing classifiers. Moreover, we provide a simple post-processing algorithm to find this solution: withholding predictive information for randomly chosen inputs to achieve parity and preserve calibration. However, this simple post-processing method is fundamentally unsatisfactory: although the post-processed predictions of our information-withholding algorithm are "fair" in expectation, most practitioners would object to the fact that a non-trivial portion of the individual predictions are withheld as a result of coin tosses – especially in sensitive settings such as health care or criminal justice. The optimality of this algorithm thus has troubling implications and shows that calibration and error-rate fairness are inherently at odds (even beyond the initial results by [8] and [26]).

Finally, we evaluate these theoretical findings empirically, comparing calibrated notions of non-discrimination against the (uncalibrated) Equalized Odds framework on several datasets. These experiments further support our conclusion that calibration and error-rate constraints are in most cases mutually incompatible goals. In practical settings, it may be advisable to choose only one of these goals rather than attempting to achieve some relaxed notion of both.

## 2   Related Work

**Calibrated probability estimates** are considered necessary for empirical risk analysis tools [4, 10, 12, 16]. In practical applications, uncalibrated probability estimates can be misleading in the sense that the end user of these estimates has an incentive to mistrust (and therefore potentially misuse) them. We note however that calibration does not remove all potential for misuse, as the end user's biases might cause her or him to treat estimates differently based on group membership. There are several post-processing methods for producing calibrated outputs from classification algorithms. For example, Platt Scaling [31] passes outputs through a learned sigmoid function, transforming them into calibrated probabilities. Histogram Binning and Isotonic Regression [35] learn a general monotonic function from outputs to probabilities. See [30] and [18] for empirical comparisons of these methods.

**Equalized Odds** [19], also referred to as *Disparate Mistreatment* [37], ensures that no error type disproportionately affects any particular group. Hardt et al. [19] provide a post-processing technique to achieve this framework, while Zafar et al. [37] introduce optimization constraints to achieve non-discrimination at training time. Recently, this framework has received significant attention from the algorithmic fairness community. Researchers have found that it is incompatible with other notions of fairness [8, 9, 26]. Additionally, Woodworth et al. [34] demonstrate that, under certain assumptions, post-processing methods for achieving non-discrimination may be suboptimal.

**Alternative fairness frameworks** exist and are continuously proposed. We highlight several of these works, though by no means offer a comprehensive list. (More thorough reviews can be found in [2, 4, 32]). It has been shown that, under most frameworks of fairness, there is a trade-off between algorithmic performance and non-discrimination [4, 9, 19, 39]. Several works approach fairness through the lens of *Statistical Parity* [6, 7, 14, 20, 22, 23, 29, 38]. Under this definition, group membership should not affect the prediction of a classifier, i.e. members of different groups should have the same probability of receiving a positive-class prediction. However, it has been argued that Statistical Parity may not be applicable in many scenarios [8, 13, 19, 26], as it attempts to guarantee equal representation. For example, it is inappropriate in criminal justice, where base rates differ across different groups. A related notion is *Disparate Impact* [15, 36], which states that the prediction rates for any two groups should not differ by more than $80\%$ (a number motivated by legal requirements). Dwork et al. [13] introduce a notion of fairness based on the idea that similar individuals should receive similar outcomes, though it challenging to achieve this notion in practice. Fairness has also been considered in online learning [21, 24], unsupervised learning [5], and causal inference [25, 27].

# 3 Problem Setup

The setup of our framework most follows the *Equalized Odds* framework [19, 37]; however, we extend their framework for use with probabilistic classifiers. Let $P \subset \mathbb{R}^k \times \{0, 1\}$ be the input space of a binary classification task. In our criminal justice example, $(\mathbf{x}, y) \sim P$ represents a person, with $\mathbf{x}$ representing the individual's history and $y$ representing whether or not the person will commit another crime. Additionally, we assume the presence of two groups $G_1, G_2 \subset P$, which represent disjoint population subsets, such as different races. We assume that the groups have different *base rates* $\mu_t$, or probabilities of belonging to the positive class: $\mu_1 = \mathrm{P}_{(\mathbf{x},y)\sim G_1}[y = 1] \neq \mathrm{P}_{(\mathbf{x},y)\sim G_2}[y = 1] = \mu_2$.

Finally, let $h_1, h_2 : \mathbb{R}^k \rightarrow [0, 1]$ be binary classifiers, where $h_1$ classifies samples from $G_1$ and $h_2$ classifies samples from $G_2$.[2] Each classifier outputs the probability that a given sample $\mathbf{x}$ belongs to the positive class. The notion of Equalized Odds non-discrimination is based on the false-positive and false-negative rates for each group, which we generalize here for use with probabilistic classifiers:

**Definition 1.** *The* generalized false-positive rate *of classifier $h_t$ for group $G_t$ is* $c_{fp}(h_t) = \mathbb{E}_{(\mathbf{x},y)\sim G_t}\big[h_t(\mathbf{x}) \mid y = 0\big]$. *Similarly, the* generalized false-negative rate *of classifier $h_t$ is* $c_{fn}(h_t) = \mathbb{E}_{(\mathbf{x},y)\sim G_t}\big[(1 - h_t(\mathbf{x})) \mid y{=}1\big]$.

If the classifier were to output either 0 or 1, this represents the standard notions of false-positive and false-negative rates. We now define the Equalized Odds framework (generalized for probabilistic classifiers), which aims to ensure that errors of a given type are not biased against any group.

**Definition 2** (Probabilistic Equalized Odds). *Classifiers $h_1$ and $h_2$ exhibit Equalized Odds for groups $G_1$ and $G_2$ if $c_{fp}(h_1) = c_{fp}(h_2)$ and $c_{fn}(h_1) = c_{fn}(h_2)$.*

**Calibration Constraints.** As stated in the introduction, these two conditions do not necessarily prevent discrimination if the classifier predictions do not represent well-calibrated probabilities. Recall that calibration intuitively says that probabilities should carry semantic meaning: if there are 100 people in $G_1$ for whom $h_1(\mathbf{x}) = 0.6$, then we expect 60 of them to belong to the positive class.

**Definition 3.** *A classifier $h_t$ is* perfectly calibrated *if $\forall p \in [0, 1]$, $\mathrm{P}_{(\mathbf{x},y)\sim G_t}\big[y{=}1 \mid h_t(\mathbf{x}){=}p\big] = p$.*

It is commonly accepted amongst practitioners that both classifiers $h_1$ and $h_2$ should be calibrated *with respect to groups $G_1$ and $G_2$* to prevent discrimination [4, 10, 12, 16]. Intuitively, this prevents the probability scores from carrying group-specific information. Unfortunately, Kleinberg et al. [26] (as well as [8], in a binary setting) prove that a classifier cannot achieve both calibration and Equalized Odds, even in an approximate sense, except in the most trivial of cases.

## 3.1 Geometric Characterization of Constraints

We now will characterize the calibration and error-rate constraints with simple geometric intuitions. Throughout the rest of this paper, all of our results can be easily derived from this interpretation. We begin by defining the region of classifiers which are *trivial*, or those that output a constant value for all inputs (i.e. $h^c(\mathbf{x}) = c$, where $0 \leq c \leq 1$ is a constant). We can visualize these classifiers on a graph with generalized false-positive rates on one axis and generalized false-negatives on the other. It follows from the definitions of generalized false-positive/false-negative rates and calibration that all trivial classifiers $h$ lie on the diagonal defined by $c_{fp}(h) + c_{fn}(h) = 1$ (Figure 1a). Therefore, all classifiers that are "better than random" must lie below this diagonal in false-positive/false-negative space (the gray triangle in the figure). Any classifier that lies above the diagonal performs "worse than random," as we can find a point on the trivial classifier diagonal with lower false-positive and false-negative rates.

Now we will characterize the set of calibrated classifiers for groups $G_1$ and $G_2$, which we denote as $\mathcal{H}_1^*$ and $\mathcal{H}_2^*$. Kleinberg et al. show that the generalized false-positive and false-negative rates of a calibrated classifier are linearly related by the base rate of the group:[3]

$$c_{fn}(h_t) = (1 - \mu_t)/\mu_t \, c_{fp}(h_t). \tag{1}$$

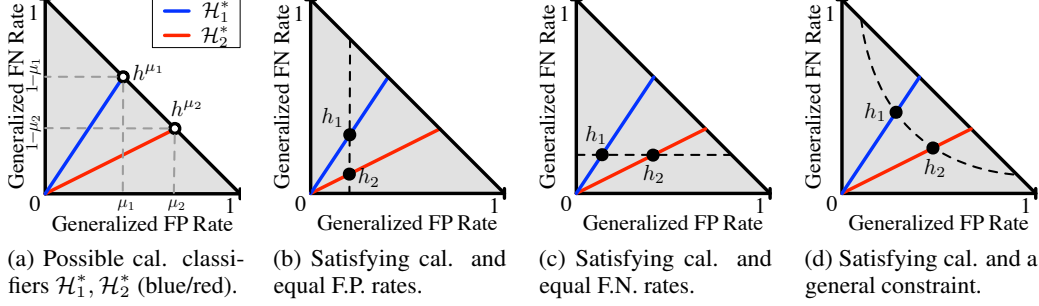

Figure 1: Calibration, trivial classifiers, and equal-cost constraints – plotted in the false-pos./false-neg. plane. $\mathcal{H}_1^*, \mathcal{H}_2^*$ are the set of cal. classifiers for the two groups, and $h^{\mu_1}, h^{\mu_2}$ are trivial classifiers.

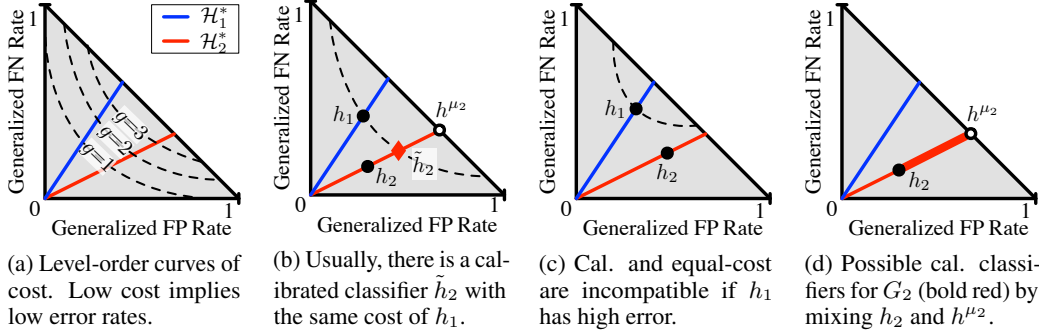

Figure 2: Calibration-Preserving Parity through interpolation.

In other words, $h_1$ lies on a line with slope $(1-\mu_1)/\mu_1$ and $h_2$ lies on a line with slope $(1-\mu_2)/\mu_2$ (Figure 1a). The lower endpoint of each line is the *perfect classifier*, which assigns the correct prediction with complete certainty to every input. The upper endpoint is a trivial classifier, as no calibrated classifier can perform "worse than random" (see Lemma 3 in Section S2). The only trivial classifier that satisfies the calibration condition for a group $G_t$ is the one that outputs the base rate $\mu_t$. We will refer to $h^{\mu_1}$ and $h^{\mu_2}$ as the trivial classifiers, calibrated for groups $G_1$ and $G_2$ respectively. It follows from the definitions that $c_{fp}(h^{\mu_1}) = \mu_1$ and $c_{fn}(h^{\mu_1}) = 1 - \mu_1$, and likewise for $h^{\mu_2}$.

Finally, it is worth noting that for calibrated classifiers, a lower false-positive rate necessarily corresponds to a lower false-negative rate and vice-versa. In other words, for a given base rate, a "better" calibrated classifier lies closer to the origin on the line of calibrated classifiers.

**Impossibility of Equalized Odds with Calibration.** With this geometric intuition, we can provide a simplified proof of the main impossibility result from [26]:

**Theorem** (Impossibility Result [26])**.** *Let $h_1$ and $h_2$ be classifiers for groups $G_1$ and $G_2$ with $\mu_1 \neq \mu_2$. $h_1$ and $h_2$ satisfy the Equalized Odds and calibration conditions if and only if $h_1$ and $h_2$ are perfect predictors.*

Intuitively, the three conditions define a set of classifiers which is overconstrained. Equalized Odds stipulates that the classifiers $h_1$ and $h_2$ must lie on the same coordinate in the false-positive/false-negative plane. As $h_1$ must lie on the blue line of calibrated classifiers for $\mathcal{H}_1^*$ and $h_2$ on the red line $\mathcal{H}_2^*$ they can only satisfy EO at the unique intersection point — the origin (and location of the perfect classifier). This implies that unless the two classifiers achieve perfect accuracy, we must relax the Equalized Odds conditions if we want to maintain calibration.

## 4 Relaxing Equalized Odds to Preserve Calibration

In this section, we show that a substantially simplified notion of Equalized Odds is compatible with calibration. We introduce a general relaxation that seeks to satisfy a *single equal-cost constraint* while maintaining calibration for each group $G_t$. We begin with the observation that Equalized

Odds sets constraints to equalize false-positives $c_{fp}(h_t)$ and false-negatives $c_{fn}(h_t)$. To capture and generalize this, we define a *cost function* $g_t$ to be a linear function in $c_{fp}(h_t)$ and $c_{fn}(h_t)$ with arbitrary dependence on the group's base rate $\mu_t$. More formally, a cost function for group $G_t$ is

$$g_t(h_t) = a_t c_{fp}(h_t) + b_t c_{fn}(h_t) \tag{2}$$

where $a_t$ and $b_t$ are non-negative constants that are specific to each group (and thus may depend on $\mu_t$): see Figure 1d. We also make the assumption that for any $\mu_t$, at least one of $a_t$ and $b_t$ is nonzero, meaning $g_t(h_t) = 0$ if and only if $c_{fp}(h_t) = c_{fn}(h_t) = 0$.[4] This class of cost functions encompasses a variety of scenarios. As an example, imagine an application in which the equal false-positive condition is essential but not the false-negative condition. Such a scenario may arise in our recidivism-prediction example, if we require that non-repeat offenders of any race are not disproportionately labeled as high risk. If we plot the set of calibrated classifiers $\mathcal{H}_1^*$ and $\mathcal{H}_2^*$ on the false-positive/false-negative plane, we can see that ensuring the false-positive condition requires finding classifiers $h_1 \in \mathcal{H}_1^*$ and $h_2 \in \mathcal{H}_2^*$ that fall on the same vertical line (Figure 1b). Conversely, if we instead choose to satisfy only the false-negative condition, we would find classifiers $h_1$ and $h_2$ that fall on the same horizontal (Figure 1c). Finally, if both false-positive and false-negative errors incur a negative cost on the individual, we may choose to equalize a weighted combination of the error rates [3, 4, 8], which can be graphically described by the classifiers lying on a convex and negatively-sloped level set (Figure 1d). With these definitions, we can formally define our relaxation:

**Definition 4** (Relaxed Equalized Odds with Calibration). *Given a cost function $g_t$ of the form in* (2)*, classifiers $h_1$ and $h_2$ achieve* Relaxed Equalized Odds with Calibration *for groups $G_1$ and $G_2$ if both classifiers are calibrated and satisfy the constraint $g_1(h_1) = g_2(h_2)$.*

It is worth noting that, for calibrated classifiers, an increase in cost strictly corresponds to an increase in both the false-negative and false-positive rate. This can be interpreted graphically, as the level-order cost curves lie further away from the origin as cost increases (Figure 2a). In other words, the cost function can always be used as a proxy for either error rate.[5]

**Feasibility.** It is easy to see that Definition 4 is always satisfiable – in Figures 1b, 1c, and 1d we see that there are many such solutions that would lie on a given level-order cost curve while maintaining calibration, including the case in which both classifiers are perfect. In practice, however, not all classifiers are achievable. For the rest of the paper, we will assume that we have access to "optimal" (but possibly discriminatory) calibrated classifiers $h_1$ and $h_2$ such that, due to whatever limitations there are on the predictability of the task, we are unable to find other classifiers that have lower cost with respect to $g_t$. We allow $h_1$ and $h_2$ to be learned in any way, as long as they are calibrated. Without loss of generality, for the remainder of the paper, we will assume that $g_1(h_1) \geq g_2(h_2)$.

Since by assumption we have no way to find a classifier for $G_1$ with lower cost than $h_1$, our goal is therefore to find a classifier $\tilde{h}_2$ with cost equal to $h_1$. This pair of classifiers would represent the lowest cost (and therefore optimal) set of classifiers that satisfies calibration and the equal cost constraint. For a given base rate $\mu_t$ and value of the cost function $g_t$, a calibrated classifier's position in the generalized false-positive/false-negative plane is uniquely determined (Figure 2a). This is because each level-order curve of the cost function $g_t$ has negative slope in this plane, and each level order curve only intersects a group's calibrated classifier line once. In other words, there is a unique solution in the false-positive/false-negative plane for classifier $\tilde{h}_2$ (Figure 2b).

Consider the range of values that $g_t$ can take. As noted above, $g_t(h_t) \geq 0$, with equality if and only if $h_t$ is the perfect classifier. On the other hand, the trivial classifier (again, which outputs the constant $\mu_t$ for all inputs) is the calibrated classifier that achieves maximum cost for any $g_t$ (see Lemma 3 in Section S2). As a result, the cost of a classifier for group $G_t$ is between 0 and $g_t(h^{\mu_t})$. This naturally leads to a characterization of feasibility: Definition 4 can be achieved if and only if $h_1$ incurs less cost than group $G_2$'s trivial classifier $h^{\mu_2}$; i.e. if $g_1(h_1) \leq g_2(h^{\mu_2})$. This can be seen graphically in Figure 2c, in which the level-order curve for $g_1(h_1)$ does not intersect the set of calibrated classifiers for $G_2$. Since, by assumption, we cannot find a calibrated classifier for $G_1$ with strictly smaller cost than $h_1$, there is no feasible solution. On the other hand, if $h_1$ incurs less cost than $h^{\mu_2}$, then we will show feasibility by construction with a simple algorithm.

**An Algorithm.** While it may be possible to encode the constraints of Definition 4 into the training procedure of $h_1$ and $h_2$, it is not immediately obvious how to do so. Even naturally probabilistic

algorithms, such as logistic regression, can become uncalibrated in the presence of optimization constraints (as is the case in [37]). It is not straightforward to encode the calibration constraint if the probabilities are assumed to be continuous, and post-processing calibration methods [31, 35] would break equal-cost constraints by modifying classifier scores. Therefore, we look to achieve the calibrated Equalized Odds relaxation by post-processing existing calibrated classifiers.

Again, given $h_1$ and $h_2$ with $g_1(h_1) \geq g_2(h_2)$, we want to arrive at a calibrated classifier $\tilde{h}_2$ for group $G_2$ such that $g_1(h_1) = g_2(\tilde{h}_2)$. Recall that, under our assumptions, this would be the best possible solution with respect to classifier cost. We show that this cost constraint can be achieved by withholding predictive information for a randomly chosen subset of group $G_2$. In other words, rather than always returning $h_2(\mathbf{x})$ for all samples, we will occasionally return the group's mean probability (i.e. the output of the trivial classifier $h^{\mu_2}$). In Lemma 4 in Section S2, we show that if

$$\tilde{h}_2(\mathbf{x}) = \begin{cases} h^{\mu_2}(\mathbf{x}) = \mu_2 & \text{with probability } \alpha \\ h_2(\mathbf{x}) & \text{with probability } 1 - \alpha \end{cases} \tag{3}$$

then the cost of $\tilde{h}_2$ is a linear interpolation between the costs of $h_2$ and $h^{\mu_2}$ (Figure 2d). More formally, we have that $g_2(\tilde{h}_2) = (1 - \alpha)g_2(h_2) + \alpha g_2(h^{\mu_2}))$, and thus setting $\alpha = \frac{g_1(h_1) - g_2(h_2)}{g_2(h^{\mu_2}) - g_2(h_2)}$ ensures that $g_2(\tilde{h}_2) = g_1(h_1)$ as desired (Figure 2b). Moreover, this randomization preserves calibration (see Section S4). Algorithm 1 summarizes this method.

---

**Algorithm 1** Achieving Calibration and an Equal-Cost Constraint via Information Withholding

> **Input:** classifiers $h_1$ and $h_2$ s.t. $g_2(h_2) \leq g_1(h_1) \leq g_2(h^{\mu_2})$, holdout set $P_{valid}$.
> - Determine base rate $\mu_2$ of $G_2$ (using $P_{valid}$) to produce trivial classifier $h^{\mu_2}$.
> - Construct $\tilde{h}_2$ using with $\alpha = \frac{g_1(h_1) - g_2(h_2)}{g_2(h^{\mu_2}) - g_2(h_2)}$, where $\alpha$ is the interpolation parameter.
>
> **return** $h_1, \tilde{h}_2$ — which are calibrated and satisfy $g_1(h_1) = g_2(\tilde{h}_2)$.

---

**Implications.** In a certain sense, Algorithm 1 is an "optimal" method because it arrives at the unique false-negative/false-positive solution for $\tilde{h}_2$, where $\tilde{h}_2$ is calibrated and has cost equal to $h_1$. Therefore (by our assumptions) we can find no better classifiers that satisfy Definition 4. This simple result has strong consequences, as the tradeoffs to satisfy both calibration and the equal-cost constraint are often unsatisfactory — both intuitively and experimentally (as we will show in Section 5).

We find two primary objections to this solution. First, it equalizes costs simply by making a classifier strictly worse for one of the groups. Second, it achieves this cost increase by withholding information on a randomly chosen population subset, making the outcome inequitable within the group (as measured by a standard measure of inequality like the Gini coefficient). Due to the optimality of the algorithm, the former of these issues is unavoidable in *any* solution that satisfies Definition 4. The latter, however, is slightly more subtle, and brings up the question of *individual fairness* (what guarantees we would like an algorithm to make with respect to each individual) and how it interacts with *group fairness* (population-level guarantees). While this certainly is an important issue for future work, in this particular setting, even if one could find another algorithm that distributes the burden of additional cost more equitably, any algorithm will make at least as many false-positive/false-negative errors as Algorithm 1, and these misclassifications will always be tragic to the individuals whom they affect. The performance loss across the entire group is often significant enough to make this combination of constraints somewhat worrying to use in practice, regardless of the algorithm.

**Impossibility of Satisfying Multiple Equal-Cost Constraints.** It is natural to argue there might be multiple cost functions that we would like to equalize across groups. However, satisfying more than one distinct equal-cost constraint (i.e. different curves in the F.P./F.N. plane) is infeasible.

**Theorem 1** (Generalized impossibility result). *Let $h_1$ and $h_2$ be calibrated classifiers for $G_1$ and $G_2$ with equal cost with respect to $g_t$. If $\mu_1 \neq \mu_2$, and if $h_1$ and $h_2$ also have equal cost with respect to a different cost function $g'_t$, then $h_1$ and $h_2$ must be perfect classifiers.*

(Proof in Section S5). Note that this is a generalization of the impossibility result of [26]. Furthermore, we show in Theorem 9 (in Section S5) that this holds in an approximate sense: if calibration and multiple distinct equal-cost constraints are approximately achieved by some classifier, then that classifier must have approximately zero generalized false-positive and false-negative rates.

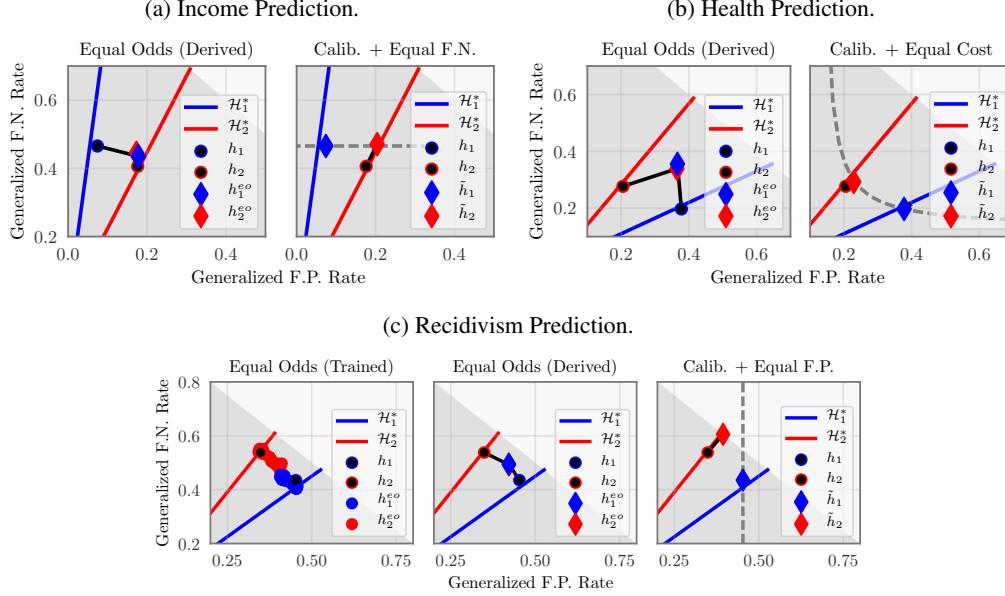

Figure 3: Generalized F.P. and F.N. rates for two groups under Equalized Odds and the calibrated relaxation. Diamonds represent post-processed classifiers. Points on the Equalized Odds (trained) graph represent classifiers achieved by modifying constraint hyperparameters.

# 5 Experiments

In light of these findings, our goal is to understand the impact of imposing calibration and an equal-cost constraint on real-world datasets. We will empirically show that, in many cases, this will result in performance degradation, while simultaneously increasing other notions of disparity. We perform experiments on three datasets: an income-prediction, a health-prediction, and a criminal recidivism dataset. For each task, we choose a cost function within our framework that is appropriate for the given scenario. We begin with two calibrated classifiers $h_1$ and $h_2$ for groups $G_1$ and $G_2$. We assume that these classifiers cannot be significantly improved without more training data or features. We then derive $\tilde{h}_2$ to equalize the costs while maintaining calibration. The original classifiers are trained on a portion of the data, and then the new classifiers are derived using a separate holdout set. To compare against the (uncalibrated) Equalized Odds framework, we derive F.P./F.N. matching classifiers using the post-processing method of [19] (**EO-Derived**). On the criminal recidivism dataset, we additionally learn classifiers that directly encode the Equalized Odds constraints, using the methods of [37] (**EO-Trained**). (See Section S6 for detailed training and post-processing procedures.) We visualize model error rates on the generalized F.P. and F.N. plane. Additionally, we plot the calibrated classifier lines for $G_1$ and $G_2$ to visualize model calibration.

**Income Prediction.** The Adult Dataset from UCI Machine Learning Repository [28] contains 14 demographic and occupational features for various people, with the goal of predicting whether a person's income is above $50,000$. In this scenario, we seek to achieve predictions with equalized cost across genders ($G_1$ represents women and $G_2$ represents men). We model a scenario where the primary concern is ensuring equal generalized F.N. rates across genders, which would, for example, help job recruiters prevent gender discrimination in the form of underestimated salaries. Thus, we choose our cost constraint to require equal generalized F.N. rates across groups. In Figure 3a, we see that the original classifiers $h_1$ and $h_2$ approximately lie on the line of calibrated classifiers. In the left plot (EO-Derived), we see that it is possible to (approximately) match both error rates of the classifiers at the cost of $h_1^{eo}$ deviating from the set of calibrated classifiers. In the right plot, we see that it is feasible to equalize the generalized F.N. rates while maintaining calibration. $h_1$ and $\tilde{h}_2$ lie on the same level-order curve of $g_t$ (represented by the dashed-gray line), and simultaneously remain on the "line" of calibrated classifiers. It is worth noting that achieving either notion of non-discrimination requires some cost to at least one of the groups. However, maintaining calibration further increases the difference in F.P. rates between groups. In some sense, the calibrated framework trades off one notion of disparity for another while simultaneously increasing the overall error rates.

**Health Prediction.** The Heart Dataset from the UCI Machine Learning Repository contains 14 processed features from 906 adults in 4 geographical locations. The goal of this dataset is to accurately predict whether or not an individual has a heart condition. In this scenario, we would like to reduce disparity between middle-aged adults ($G_1$) and seniors ($G_2$). In this scenario, we consider F.P. and F.N. to both be undesirable. A false prediction of a heart condition could result in unnecessary medical attention, while false negatives incur cost from delayed treatment. We therefore utilize the following cost function $g_t(h_t) = r_{fp} h_t(\mathbf{x}) (1 - y) + r_{fn} (1 - h_t(\mathbf{x})) y$, which essentially assigns a weight to both F.N. and F.P. predictions. In our experiments, we set $r_{fp} = 1$ and $r_{fn} = 3$. In the right plot of Figure 3b, we can see that the level-order curves of the cost function form a curved line in the generalized F.P./F.N. plane. Because our original classifiers lie approximately on the same level-order curve, little change is required to equalize the costs of $h_1$ and $h_2$ while maintaining calibration. This is the only experiment in which the calibrated framework incurs little additional cost, and therefore could be considered a viable option. However, it is worth noting that, in this example, the equal-cost constraint does not explicitly match either of the error types, and therefore the two groups will in expectation experience different types of errors. In the left plot of Figure 3b (EO-Derived), we see that it is alternatively feasible to explicitly match both the F.P. and F.N. rates while sacrificing calibration.

**Criminal Recidivism Prediction.** Finally, we examine the frameworks in the context of our motivating example: criminal recidivism. As mentioned in the introduction, African Americans ($G_1$) receive a disproportionate number of F.P. predictions as compared with Caucasians ($G_2$) when automated risk tools are used in practice. Therefore, we aim to equalize the generalized F.P. rate. In this experiment, we modify the predictions made by the COMPAS tool [12], a risk-assessment tool used in practice by the American legal system. Additionally, we also see if it is possible to improve the classifiers with training-time Equalized Odds constraints using the methods of Zafar et al. [37] (EO-Trained). In Figure 3c, we first observe that the original classifiers $h_1$ and $h_2$ have large generalized F.P. and F.N. rates. Both methods of achieving Equalized Odds — training constraints (left plot) and post-processing (middle plot) match the error rates while sacrificing calibration. However, we observe that, assuming $h_1$ and $h_2$ cannot be improved, it is infeasible to achieve the calibrated relaxation (Figure 3c right). This is an example where matching the F.P. rate of $h_1$ would require a classifier worse than the trivial classifier $h^{\mu_2}$. This example therefore represents an instance in which calibration is completely incompatible with any error-rate constraints. If the primary concern of criminal justice practitioners is calibration [12, 16], then there will inherently be discrimination in the form of F.P. and F.N. rates. However, if the Equalized Odds framework is adopted, the miscalibrated risk scores inherently cause discrimination to one group, as argued in the introduction. Therefore, the most meaningful change in such a setting would be an improvement to $h_2$ (the classifier for African Americans) either through the collection of more data or the use of more salient features. A reduction in overall error to the group with higher cost will naturally lead to less error-rate disparity.

# 6 Discussion and Conclusion

We have observed cases in which calibration and relaxed Equalized Odds are compatible and cases where they are not. When it is feasible, the penalty of equalizing cost is amplified if the base rates between groups differ significantly. This is expected, as base rate differences are what give rise to cost-disparity in the calibrated setting. Seeking equality with respect to a single error rate (e.g. false-negatives, as in the income prediction experiment) will necessarily increase disparity with respect to the other error. This may be tolerable (in the income prediction case, some employees will end up over-paid) but could also be highly problematic (e.g. in criminal justice settings). Finally, we have observed that the calibrated relaxation is infeasible when the best (discriminatory) classifiers are not far from the trivial classifiers (leaving little room for interpolation). In such settings, we see that calibration is completely incompatible with an equalized error constraint.

In summary, we conclude that maintaining cost parity *and* calibration is desirable yet often difficult in practice. Although we provide an algorithm to effectively find the unique feasible solution to both constraints, it is inherently based on randomly exchanging the predictions of the better classifier with the trivial base rate. Even if fairness is reached in expectation, for an individual case, it may be hard to accept that occasionally consequential decisions are made by randomly withholding predictive information, irrespective of a particular person's feature representation. In this paper we argue that, as long as calibration is required, no lower-error solution can be achieved.

## Acknowledgements

GP, FW, and KQW are supported in part by grants from the National Science Foundation (III-1149882, III-1525919, III-1550179, III-1618134, and III-1740822), the Office of Naval Research DOD (N00014-17-1-2175), and the Bill and Melinda Gates Foundation. MR is supported by an NSF Graduate Research Fellowship (DGE-1650441). JK is supported in part by a Simons Investigator Award, an ARO MURI grant, a Google Research Grant, and a Facebook Faculty Research Grant.

## Footnotes

*Equal contribution, alphebetical order.

[1] For the remainder of the paper, we will use *Equalized Odds* to refer to this notion of non-discrimination.

[2] In practice, $h_1$ and $h_2$ can be trained jointly (i.e. they are the same classifier).

[3] Throughout this work we will treat the calibration constraint as holding exactly; however, our results generalize to approximate settings as well. See the Supplementary Materials for more details.

[4] By calibration, we cannot have one of $c_{fp}(h_t) = 0$ or $c_{fn}(h_t) = 0$ without the other, see Figure 1a.

[5] This holds even for approximately calibrated classifiers — see Section S3.

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
