[Reviews · NeurIPS 2017]

Reviewer 1



The paper deals with an increasingly important topic of fairness in predictive analytics. The focus of the paper is on studying analytically and experimentally the model calibration effect on fairness. The paper presentation is excellent. The problem is well presented and positioned with respect to existing work. The authors characterize the calibration and error-rate constraints with simple geometric intuitions and link theoretical findings to that interpretation. It is an interesting result that it is difficult to enforce calibration and there are conditions under which it is feasible and a unique optimal solution exists and can be found efficiently through post-processing of existing classifiers. In the experimental study the authors study the impact of imposing calibration and an equal cost constraint on real-world datasets. The results are promising.

Reviewer 2



This paper explores the relationship between Equal Opportunity and calibration constraints on models. They show that even with relaxed Equal Opportunity constraints it can be problematic to enforce calibration. They provide necessary and sufficient conditions for achieving this. They also create an algorithm to achieve a relaxation on Equal Opportunity that requires only one constraint (False positive or false negative or a weighted sum) be met by withholding information. They also demonstrated the effect of this algorithm on several relevant datasets. I found this paper to be well-written, easy to follow and also very thorough. I think the topic is an important one for machine learning in practice, and the paper definitely adds something to the discussion. I appreciated both the work on showing the required conditions and as well as the algorithm, which helps to understand the effect of this in practice.

Reviewer 3



Note: Equal opportunity as used in this paper was originally defined as “Equalized odds” in Hardt et al. 2017. This needs to be fixed in multiple places. The paper analyzes the interaction between relaxations of equalized odds criteria and classifier calibration. The work extends and builds on the results of a previous work [23]. 1. Incompatibility of calibration and non-discrimination: The previous work [23] already establish incompatibility of matching both FPR and FNR (equalized odds) and calibration in exact as well as approximate sense (except in trivial classifiers). This result has been extended to satisfying non-discrimination with respect to two generalized cost functions expressed as a linear combinations of FPR and TPR of respective groups. This generalization while new is not entirely surprising . 2. The authors show that if the equalized odds is relaxed to non-discrimination with respect to a single cost function expressed as a linear combination of FPR and FNR, then the relaxation and calibration can be jointly achieved under certain feasibility conditions. 3. Additional experiments are provided to compare the effects of enforcing non-discrimination and/or calibration. While there are some new results and additional experiments, the conclusions are fairly similar to [23], essentially establishing incompatibility of satisfying equalized-odds style non-discrimination and classification calibration. Additional comments: 1. Why was EO trained not implemented and compared for the heart disease and income datasets? 2. Line 165-166: there seems to be a missing explanation or incorrect assumption here as if only one of a_t or b_t is required to be nonzero then g_t can be zero with just FPR=0 OR FNR=0, respectively. [Post rebuttal] Thanks for the clarifications on the contributions and the condition on a_t,b_t. I see the significance of the quantification of degradation performance of classifiers even when training is mindful of fairness and calibration conditions -- a similar quantification of accuracy of "best" fair predictor for exclusively EO condition (without calibration) would form a good complementary result (I dont expect the authors to attempt it for this paper, but may be as future work). That said, given the existing literature, I am still not very surprised by the other conclusions on incompatibility of two fairness conditions with calibration.